# Association of primary and community care services with emergency visits and hospital admissions at the end of life in people with cancer: a retrospective cohort study

Javiera Leniz [ID],[1] Lesley A Henson,[1] Jean Potter,[2] Wei Gao [ID],[1] Tom Newsom-Davis,[3] Zia Ul-Haq,[4] Amanda Lucas,[4] Irene J Higginson,[1] Katherine E Sleeman[1]

[1]Cicely Saunders Institute of Palliative Care, Policy and Rehabilitation, King's College London, London, UK
[2]Department of Palliative Care, Hillingdon Hospitals NHS Foundation Trust, Uxbridge, UK
[3]Oncology, Chelsea and Westminster Hospital NHS Foundation Trust, London, UK
[4]Discover-Now, Imperial College Health Partners, London, UK

**Correspondence to**
Javiera Leniz;
javiera.martelli@kcl.ac.uk

## ABSTRACT

**Objective** To examine the association between primary and community care use and measures of acute hospital use in people with cancer at the end of life.

**Design** Retrospective cohort study.

**Setting** We used Discover, a linked administrative and clinical data set from general practices, community and hospital records in North West London (UK).

**Participants** People registered in general practices, with a diagnosis of cancer who died between 2016 and 2019.

**Primary and secondary outcome measures** ≥3 hospital admissions during the last 90 days, ≥1 admissions in the last 30 days and ≥1 emergency department (ED) visit in the last 2 weeks of life.

**Results** Of 3581 people, 490 (13.7%) had ≥3 admissions in last 90 days, 1640 (45.8%) had ≥1 admission in the last 30 days, 1042 (28.6%) had ≥1 ED visits in the last 2 weeks; 1069 (29.9%) had more than one of these indicators. Contacts with community nurses in the last 3 months (≥13 vs <4) were associated with fewer admissions in the last 30 days (risk ratio (RR) 0.88, 95% CI 0.90 to 0.98) and ED visits in the last 2 weeks of life (RR 0.79, 95% CI 0.68 to 0.92). Contacts with general practitioners in the last 3 months (≥11 vs <4) was associated with higher risk of ≥3 admissions in the last 90 days (RR 1.63, 95% CI 1.33 to 1.99) and ED visits in the last 2 weeks of life (RR 1.27, 95% CI 1.10 to 1.47).

**Conclusions** Expanding community nursing could reduce acute hospital use at the end of life and improve quality of care.

## BACKGROUND

While mortality rates for most types of cancer have decreased, globally deaths from cancer increased by 25.4% between 2007 and 2017 due to population ageing and growth.[1] A similar pattern is observed in the UK, and more than 95 000 deaths due to cancer are projected for 2035, 24.5% more than in 2014.[2] It is therefore critical to understand

## Strengths and limitations of this study

► Population-based cohort study using a large and comprehensive data set that holds information on healthcare services use from eight different boroughs in London and over 2 million people.

► Our study examined data from local authorities and general practice records, which provides a unique opportunity to describe community and primary care service utilisation.

► People in the cohort might have not died from cancer but from other conditions, as information on cause of death was not available.

► The overall use of the primary care practice and palliative care community teams are likely to be underestimated in this study due to the methods used to estimate contacts.

► Information on the quality of care or the appropriateness of hospital admissions and emergency department visits was not available and are likely to be confounders.

how to provide high-quality end-of-life care for people with cancer.

Excessive use of hospital care in the last months of life has been proposed as an indicator of the quality of end-of-life cancer care.[3] This is because emergency hospital care is associated with reduced quality of life and care satisfaction in cancer patients and their families,[4–6] without contributing to an improvement in survival.[7 8] Despite these negative outcomes, hospital admissions and emergency department (ED) visits at the end of life have increased over time in cancer patients nearing the end of their life.[9]

Sociodemographic and illness-related factors have been found to be associated with higher hospital admissions and ED

attendances in the last months of life, for example being male, black, having lung cancer and low socioeconomic status.[10–12] On the other hand, access to palliative care services has been associated with lower acute end-of-life care, suggesting that these services could help prevent unnecessary admissions to hospital.[10 13] However, little is known about how contacts with general practitioners (GPs) in primary care practices and other community services such as community nurses, community palliative care or rehabilitation teams influence acute care use near the end of life among people with cancer. The aim of this study was to describe the association between primary and community care services use with three measures of acute hospital use for people with cancer at the end of life.

## METHODS
### Design and data sources
This is a retrospective cohort study using the Discover dataset, one of Europe's largest linked longitudinal deidentified datasets that includes 95% of all patients registered with a GP in North West London.[14] The Discover dataset is a platform that enables researcher access to pseudonymised patient-level data drawn from the Whole Systems Integrated Care local data warehouse for research purposes. Discover dataset is maintained and interrogated on a secure server and extracts of data are then aggregated in compliance with the Information Governance suppression rule where numbers below five are annotated as <5. In this process, the deidentified data are rendered anonymised by stripping out any information that would allow reidentification of an individual's identity. Discover dataset is accessible via Discover-NOW Health Data Research Hub for Real World Evidence through their data scientist specialists and IG committee-approved analysts, hosted by Imperial College Health Partners.

In June 2019, the database held records for a total of 2.37 million patients spread across eight Clinical Commissioning Groups. The estimated total population for the eight boroughs contributing data to Discover was 2.1 million in mid-2019. Differences in the population estimated and the number of patients in the data set could be explained by people being enrolled in a GP practice contributing to the dataset but whose usual place of residence is in another area. Of 370 health and social care provider organisations from the National Health Service (NHS) in North West London, 359 (97%) have a data sharing agreement and submit their records to the dataset. Organisations feeding records to the dataset include primary care practices, mental health, community trusts and hospital care attended by North West London patients, and exclude private and third sector providers, such as hospices services.[14] We chose the Discover dataset as it is a comprehensive population-based dataset and provides access to community care records in addition to primary and hospital care records, which are not generally available for other primary care datasets in the UK.

The age and gender distribution and prevalence of long-term conditions of the Discover population are similar to the rest of London and the UK.[14]

### Population
Adults (aged 18 or over) included in the Discover dataset with at least one record of a cancer diagnosis recorded at any point from 1st January 2015 onwards in primary care practice or hospital inpatient records using Read Codes and International Statistical Classification of Diseases and Related Health Problems (ICD) 10 codes, respectively (codes available in online supplemental box S1). We included in the cohort people who died between 2016 and 2019 based on the date of death recorded in primary care or hospital records. As we did not have information on the cause of death or cancer severity, we restricted our sample to people who had been identified as having palliative care needs in primary care records at any time based on the quality of outcomes framework (QoF) Read Codes for the Palliative Care register,[15] to include people whose death could be considered expected rather than sudden (codes available in online supplemental box S1).

### Outcomes
We evaluated three measures of acute hospital use towards the end of life. We chose these three outcome measures as their prevalence at a population-level can be considered an indicator of end-of-life care quality according to Henson *et al* systematic review,[3] their focus on acute care use at the end of life and the feasibility to be measured in the data. The three measures were as:
1. Three or more emergency hospital admissions in the last 90 days of life.[16]
2. One or more emergency hospital admissions in the last 30 days of life.[3]
3. One or more ED visits in the last 2 weeks of life.[3]

### Explanatory variables
▶ Primary care practice contacts: we identified contacts with the primary care practice in the last 90 days of life using a similar approach reported by Kontopantelis *et al*.[17] We considered only direct consultations such as telephone, face-to-face or home visits and excluded administrative consultations or non-attended appointments. It was not possible to identify whether the contact in the practice was with a doctor or another healthcare professional. Only one consultation in the same day was used to reduce the likelihood of including duplicate records, as it was not possible to determine whether records from the same day correspond to more than one contact or not. This approach has been widely used in research using primary care records in the UK[17 18] (online supplemental box S2)
▶ Contacts with other community services: we identified contacts with community nurses, community palliative care teams and rehabilitation teams in the last 90 days of life based on the date of the contact and the description of the service. Contacts with rehabilitation

teams included physiotherapy, speech and language and occupational therapy services. We removed non-attendant contacts and duplicates based on the date. We identified individuals who were defined by Discover primary care data set as living in a care home based on the latest patient record (online supplemental box S3).

## Co-variables

► Sociodemographic: age at death, gender, ethnicity and Index of Multiple Deprivation (IMD) were extracted from Discover dataset records for each individual. The 2015 IMD was derived at Lower Super Output Areas from patients' last address registered in the system and reported according to The English Indices of Deprivation 2015 guidance.[19]

► Illness-related: the number of comorbidities was calculated using the count of 15 QoF chronic diseases (excluding cancer) identified from Read codes in the primary care practice records.[15] The type of cancer was identified from the primary care practice and hospital in-patient records using Read Codes and ICD-10 codes, respectively (online supplemental box S1). Only 6% of the cohort had more than one cancer recorded and were included in the 'Other' category.

► Number of days in hospital: we calculated the number of days patients spent in hospital in the last 90 days of life using in-patient hospital codes for spells' start dates and discharge dates.

## Analysis

Data were described using count and percentage for categorical variables and mean and SD for continuous variables. A Pearson's $\chi^2$ test for the trend for categorical variables and t-test and Wilcoxon rank-sum test for age and days in hospital, respectively, was used to evaluate the association between each variable and the outcomes.

We used generalised estimating equations to estimate the unadjusted and multivariate association between sociodemographic, illness-related factors and contacts with primary and community care services in the last 90 days of life and each of the three indicators separately. We used Poisson family with log link function, exchangeable correlation structure and robust error variance with data clustered in primary care practices where patients were registered. For the multivariate model, we adjusted by age, gender, IMD quintile, care home residence, type of cancer and number of QoF comorbidities. We selected specific comorbidities, and primary and community care services use according to significance in unadjusted analysis ($p \leq 0.05$). We excluded ethnicity from the final model to avoid biasing the sample as the variable has a large proportion of missing data. To facilitate interpretation, we categorised primary and community care service contacts based on clinical judgement. Categories were approximately one or fewer contacts per month, more than one contact per month but less than one contact per week, and more than one contact per week, depending

on the distribution. Because number of contacts with palliative care and rehabilitation teams were small, we adapted these categories. We included the number of days each person spent in hospital in the last 90 days of life as a continuous variable in the models to account for the fact that if someone is in hospital, they cannot receive care in the community.

We performed four sensitivity analysis: (1) to explore the influence of days in hospital by removing the variable from the model, (2) to understand the impact of categorisation of primary and community care services in the model, we used the same model but with the corresponding primary and community care service use variables as continuous, (3) to explore the impact of restricting the sample to people with a record of cancer diagnosis and identification of palliative care needs in the last 12 months of life (instead of at any time) and (4) to understand the influence of ethnicity in the model.

## Patient and public involvement

The protocol was presented and discussed with patients and public representatives at the beginning of the study. A member of the public with experience caring for a relative who died with cancer joined the Project Advisory Group of the project, reviewing a lay version of the protocol and participated in the interpretation of results.

## RESULTS

### Characteristics of the cohort

We identified 4933 people with a diagnosis of cancer and who died between 2016 and 2019; 3848 (78.0%) of them had a palliative care QoF record in primary care records. After removing 267 people with invalid dates of death and hospital admissions, 3581 people were included in the analysis (online supplemental figure S1). The mean age was 76.6 (SD 13.3), 55.4% were male and 21.3% had four or more comorbidities. The most frequent cancer diagnosis was lung cancer (21.5%) followed by bowel (11.6%) and prostate cancer (8.6%) (table 1).

Of the 3581 people in the sample, 490 (13.7%) had three or more emergency admissions in last 90 days, 1640 (45.8%) had one or more emergency admissions in the last 30 days and 1042 (28.6%) had one or more ED visits in the last 2 weeks of life (table 1). There was overlap between the three indicators with 1069 (29.9%) of the sample having more than one of the indicators and 269 (7.5%) of the cohort having all three. Older age, white ethnicity and living in a care home were associated with lower chances of all three outcomes (table 1).

### Primary care and other community services

On average, people in the cohort had 2.3 (SD 3.3) telephone and 2.4 (SD 3.3) face-to-face consultations with the primary care practice, 8.6 (SD 13.7) contacts with community nurses, 1.2 (SD 3.0) contacts with community palliative care teams and 0.3 (SD 1.2) contacts with rehabilitation services in the last 90 days of life.

**Table 1** Sample characteristics by outcome measure

| | | All sample | | 3 or more EHA last 3 months | | | | | 1 or more EHA last month | | | | | 1 or more ED visit in the last 2 weeks of life | | | | |
|---|---|---|---|---|---|---|---|---|---|---|---|---|---|---|---|---|---|---|
| | | | | No | | Yes | | | No | | Yes | | | No | | Yes | | |
| | | n=3581 | | n=3091 | | n=490 | | P value | n=1941 | | n=1640 | | P value | n=2539 | | n=1042 | | P value |
| | | N | % | N | % | N | % | | N | % | N | % | | N | % | N | % | |
| Age | (Mean, SD) | 76.61 | 13.32 | 77.32 | 13.06 | 72.18 | 14.08 | <0.001 | 77.63 | 12.92 | 75.41 | 13.69 | <0.001 | 77.21 | 13.31 | 75.16 | 13.26 | <0.001 |
| Gender | Female | 1596 | 44.6 | 1394 | 45.1 | 202 | 41.2 | 0.109 | 905 | 46.6 | 691 | 42.1 | 0.007 | 1155 | 45.5 | 441 | 42.3 | 0.083 |
| | Male | 1985 | 55.4 | 1697 | 54.9 | 288 | 58.8 | | 1036 | 53.4 | 949 | 57.9 | | 1384 | 54.5 | 601 | 57.7 | |
| Ethnicity | White | 1511 | 42.2 | 1330 | 43.0 | 181 | 36.9 | 0.001 | 866 | 44.6 | 645 | 39.3 | 0.006 | 1114 | 43.9 | 397 | 38.1 | <0.001 |
| | Black | 232 | 6.5 | 195 | 6.3 | 37 | 7.6 | | 115 | 5.9 | 117 | 7.1 | | 147 | 5.8 | 85 | 8.2 | |
| | Asian | 490 | 13.7 | 396 | 12.8 | 94 | 19.2 | | 240 | 12.4 | 250 | 15.2 | | 313 | 12.3 | 177 | 17.0 | |
| | Mixed | 523 | 14.6 | 447 | 14.5 | 76 | 15.5 | | 270 | 13.9 | 253 | 15.4 | | 380 | 15.0 | 143 | 13.7 | |
| | Other | 177 | 4.9 | 149 | 4.8 | 28 | 5.7 | | 104 | 5.4 | 73 | 4.5 | | 127 | 5.0 | 50 | 4.8 | |
| | Missing | 648 | 18.1 | 574 | 18.6 | 74 | 15.1 | | 346 | 17.8 | 302 | 18.4 | | 458 | 18.0 | 190 | 18.2 | |
| IMD quintile | 1 (most deprived) | 604 | 16.9 | 499 | 16.1 | 105 | 21.4 | 0.047 | 319 | 16.4 | 285 | 17.4 | 0.613 | 414 | 16.3 | 190 | 18.2 | 0.536 |
| | 2 | 1135 | 31.7 | 983 | 31.8 | 152 | 31.0 | | 612 | 31.5 | 523 | 31.9 | | 804 | 31.7 | 331 | 31.8 | |
| | 3 | 894 | 25.0 | 776 | 25.1 | 118 | 24.1 | | 474 | 24.4 | 420 | 25.6 | | 642 | 25.3 | 252 | 24.2 | |
| | 4 | 540 | 15.1 | 467 | 15.1 | 73 | 14.9 | | 310 | 16.0 | 230 | 14.0 | | 383 | 15.1 | 157 | 15.1 | |
| | 5 (least deprived) | 299 | 8.3 | 268 | 8.7 | 31 | 6.3 | | 167 | 8.6 | 132 | 8.0 | | 222 | 8.7 | 77 | 7.4 | |
| | Missing | 109 | 3.0 | 98 | 3.2 | 11 | 2.2 | | 59 | 3.0 | 50 | 3.0 | | 74 | 2.9 | 35 | 3.4 | |
| Lived in care home | No | 3389 | 94.6 | 2910 | 94.1 | 479 | 97.8 | 0.001 | 1797 | 92.6 | 1592 | 97.1 | <0.001 | 2387 | 94.0 | 1002 | 96.2 | 0.010 |
| | Yes | 192 | 5.4 | 181 | 5.9 | 11 | 2.2 | | 144 | 7.4 | 48 | 2.9 | | 152 | 6.0 | 40 | 3.8 | |
| Number QoF comorbidities | 0 | 579 | 16.2 | 489 | 15.8 | 90 | 18.4 | 0.184 | 304 | 15.7 | 275 | 16.8 | 0.041 | 398 | 15.7 | 181 | 17.4 | 0.036 |
| | 1 | 856 | 23.9 | 750 | 24.3 | 106 | 21.6 | | 495 | 25.5 | 361 | 22.0 | | 641 | 25.2 | 215 | 20.6 | |
| | 2 | 756 | 21.1 | 657 | 21.3 | 99 | 20.2 | | 401 | 20.7 | 355 | 21.6 | | 534 | 21.0 | 222 | 21.3 | |
| | 3 | 628 | 17.5 | 529 | 17.1 | 99 | 20.2 | | 353 | 18.2 | 275 | 16.8 | | 445 | 17.5 | 183 | 17.6 | |
| | ≥4 | 762 | 21.3 | 666 | 21.5 | 96 | 19.6 | | 388 | 20.0 | 374 | 22.8 | | 521 | 20.5 | 241 | 23.1 | |

Continued

**Table 1** Continued

| | | All sample | | 3 or more EHA last 3 months | | | | | 1 or more EHA last month | | | | | 1 or more ED visit in the last 2 weeks of life | | | | |
|---|---|---|---|---|---|---|---|---|---|---|---|---|---|---|---|---|---|---|
| | | n=3581 | | No n=3091 | | Yes n=490 | | | No n=1941 | | Yes n=1640 | | | No n=2539 | | Yes n=1042 | | |
| | | N | % | N | % | N | % | P value | N | % | N | % | P value | N | % | N | % | P value |
| QoF comorbidities | COPD (yes) | 535 | 14.9 | 456 | 14.8 | 79 | 16.1 | 0.429 | 266 | 13.7 | 269 | 16.4 | 0.024 | 371 | 14.6 | 164 | 15.7 | 0.390 |
| | Depression (yes) | 625 | 17.5 | 539 | 17.4 | 86 | 17.6 | 0.951 | 330 | 17.0 | 295 | 18.0 | 0.439 | 446 | 17.6 | 179 | 17.2 | 0.781 |
| | Diabetes (yes) | 996 | 27.8 | 855 | 27.7 | 141 | 28.8 | 0.609 | 522 | 26.9 | 474 | 28.9 | 0.181 | 682 | 26.9 | 314 | 30.1 | 0.047 |
| | Hypertension (yes) | 2022 | 56.5 | 1753 | 56.7 | 269 | 54.9 | 0.452 | 1101 | 56.7 | 921 | 56.2 | 0.734 | 1441 | 56.8 | 581 | 55.8 | 0.585 |
| | Dementia (yes) | 328 | 9.2 | 300 | 9.7 | 28 | 5.7 | 0.004 | 202 | 10.4 | 126 | 7.7 | 0.005 | 246 | 9.7 | 82 | 7.9 | 0.086 |
| | CHD (yes) | 689 | 19.2 | 596 | 19.3 | 93 | 19.0 | 0.875 | 354 | 18.2 | 335 | 20.4 | 0.098 | 463 | 18.2 | 226 | 21.7 | 0.017 |
| Type of cancer | Bowel | 416 | 11.6 | 374 | 12.1 | 42 | 8.6 | 0.075 | 234 | 12.1 | 182 | 11.1 | 0.215 | 288 | 11.3 | 128 | 12.3 | 0.032 |
| | Lung | 769 | 21.5 | 646 | 20.9 | 123 | 25.1 | | 392 | 20.2 | 377 | 23.0 | | 520 | 20.5 | 249 | 23.9 | |
| | Prostate | 309 | 8.6 | 260 | 8.4 | 49 | 10.0 | | 169 | 8.7 | 140 | 8.5 | | 223 | 8.8 | 86 | 8.3 | |
| | Breast | 237 | 6.6 | 205 | 6.6 | 32 | 6.5 | | 117 | 6.0 | 120 | 7.3 | | 155 | 6.1 | 82 | 7.9 | |
| | Pancreas | 194 | 5.4 | 167 | 5.4 | 27 | 5.5 | | 103 | 5.3 | 91 | 5.5 | | 142 | 5.6 | 52 | 5.0 | |
| | Haematological | 137 | 3.8 | 114 | 3.7 | 23 | 4.7 | | 75 | 3.9 | 62 | 3.8 | | 95 | 3.7 | 42 | 4.0 | |
| | Other | 1519 | 42.4 | 1325 | 42.9 | 194 | 39.6 | | 851 | 43.8 | 668 | 40.7 | | 1116 | 44.0 | 403 | 38.7 | |

CHD, chronic heart disease; COPD, chronic obstructive pulmonary disease; ED, emergency department visits; EHA, emergency hospital admissions; IMD, Index of Multiple Deprivation; QoF, quality of outcomes framework.

People with three or more hospital admissions in the last 90 days and ED visits in the last 2 weeks of life also had more contacts in the primary care practice. Conversely, people with hospital admissions in the last 30 days and ED visits in the last 2 weeks of life had fewer contacts with community nurses and palliative care teams (table 2).

## Multivariate analysis

In the multivariate analysis, people with three or more hospital admissions in the last 90 days were more likely to be younger, have lung or prostate cancer, have more than 11 contacts with the primary care practice in the last 90 days and were less likely to live in a care home. People with one or more admissions in the last 30 days, were more likely to be younger, male, have breast cancer, they were less likely to live in a care home, and less likely to have more than 13 contacts with community nurses in the last 90 days. Having one or more ED visits in the last 2 weeks of life was associated with being younger, having fewer contacts with community nurses, more contacts with the primary care practice and lower chances of living in a care home (table 3).

The sensitivity analyses for the outcomes three or more hospital admissions in the last 90 days and ED visits in the last 2 weeks of life demonstrated similar results (online supplemental tables S1 and S3). In the sensitivity analyses for one or more admissions in the last 30 days, the number of contacts with community nurses analysed as a continuous variable was no longer associated with the outcome measure, suggesting a threshold response (online supplemental table S2).

## DISCUSSION

The three outcome measures for end-of-life cancer care evaluated (three or more admissions to hospital in the last 90 days, one or more hospital admissions in the last 30 days and one or more ED visits in the last 2 weeks of life) were frequent (13.7%, 45.8% and 28.6%, respectively). We found that contacts with community nurses were associated with fewer ED visits in the last 2 weeks of life, and contacts with the primary care practice were associated with higher risk of multiple admissions to hospital in the last 90 days and ED visits in the last 2 weeks of life.

We found contacts with community nurses were associated with a lower risk of hospital admissions and ED visits at the end of life. These findings are consistent with previous studies where more community nursing hours per week were associated with lower odds of hospital admissions and ED visits at the end of life among patients with cancer in Canada.[20 21] Community nurses have an important role providing physical care, managing symptoms and medications, educating and giving information to patients and families, and coordinating care[22–25] and therefore, they could play an important role in avoiding unnecessary hospital admissions at the end of life in people with cancer.

People with cancer who were living in care homes before death had a lower risk of hospital admissions and ED visits at the end of life in this cohort. The number of people living in care homes in our sample was small, and we did not have information on the level of healthcare needs of this group of people. Nevertheless, these findings are consistent with other international studies showing that people living in long-term facilities are less likely to have transitions to hospital regardless of the cause of death.[26 27] Long-term facilities are one of the very few care settings in the community providing continuous care including out-of-hours.[28 29] More research is needed to understand the mechanisms that explain this association, as well as to explore differences in healthcare provision and healthcare needs between people living in care homes and in the community.

In contrast with previous studies,[20 30] we found that contacts with the primary care practice were associated with higher risk of multiple admissions to hospital in the last 90 days and ED visits in the last 2 weeks of life. This is likely to be explained by complexity of healthcare needs: more severe and complex patients are likely to have a higher use of healthcare services.[31] A rise in the number of contacts with the practice could be an opportunity to identify patients who are deteriorating or whose healthcare needs are increasing. High healthcare use can also be an indicator of unmet needs, ineffective and uncoordinated care and lead to poor patient satisfaction.[32–34] Having many different healthcare professionals could cause confusion among patients and their caregivers and lead to more consultations. It is possible that more contacts with the primary care practice in this sample reflects poor coordination or a lack of continuity of care, leading to more admissions to hospital.

### Implications for research and/or practice

Primary care physicians play a key role in providing care for people approaching the end of life. Their involvement is valued by patients and families,[35] and has shown to improve end-of-life care outcomes.[36 37] However, several barriers to palliative care in general practice have been identified, such as the increasing workload and time, lack of funding, poor communication with specialists and lack of experience and training.[38 39] More research is needed to explore effective models of end-of-life care in primary care and palliative care integration in order to address the increasing demand for care and complexity of healthcare needs that patients experience when approaching the end of life.

The three measures used in this study have been proposed as quality indicators for cancer end-of-life care.[3 40] Measuring the quality of care provided by healthcare services is key to monitor and promote the delivery of high-quality cancer care. We found an overlap between these indicators, with 29.9% patients having more than one and different predictors associated with each of them, which suggests a balanced combination of quality indicators might be needed to measure the quality of care provided by healthcare services. Although the measures chosen are recognised quality indicators and

**Table 2** Healthcare services utilisation in the last 3 months of life by outcome measure

| | 3 or more EHA in the last 3 months | | | | | 1 or more EHA in the last month | | | | | 1 or more ED visits in the last 2 weeks | | | | |
| | No | | Yes | | P value* | No | | Yes | | P value* | No | | Yes | | P value* |
| | N | % | N | % | | N | % | N | % | | N | % | N | % | |
| **Contacts with GP practice** | | | | | | | | | | | | | | | |
| 0–3 | 1812 | 58.6 | 243 | 49.6 | | 1144 | 58.9 | 911 | 55.5 | | 1499 | 59.0 | 556 | 53.4 | |
| 4–10 | 873 | 28.2 | 149 | 30.4 | | 539 | 27.8 | 483 | 29.5 | | 705 | 27.8 | 317 | 30.4 | |
| ≥11 | 406 | 13.1 | 98 | 20.0 | <0.001 | 258 | 13.3 | 246 | 15.0 | 0.105 | 355 | 14.0 | 169 | 16.2 | 0.014 |
| **Contacts with community nurses** | | | | | | | | | | | | | | | |
| 0–3 | 1629 | 52.7 | 244 | 49.8 | | 995 | 51.3 | 878 | 53.5 | | 1300 | 51.2 | 573 | 55.0 | |
| 4–12 | 712 | 23.0 | 123 | 25.1 | | 420 | 21.6 | 415 | 25.3 | | 582 | 22.9 | 253 | 24.3 | |
| ≥13 | 725 | 23.5 | 117 | 23.9 | 0.475 | 504 | 26.0 | 338 | 20.6 | <0.001 | 632 | 24.9 | 210 | 20.2 | 0.008 |
| **Contacts with community palliative care teams** | | | | | | | | | | | | | | | |
| 0–3 | 2652 | 85.8 | 432 | 88.2 | | 1644 | 84.7 | 1440 | 87.8 | | 2170 | 85.5 | 914 | 87.7 | |
| 4–8 | 313 | 10.1 | 40 | 8.2 | | 206 | 10.6 | 147 | 9.0 | | 254 | 10.0 | 99 | 9.5 | |
| ≥9 | 126 | 4.1 | 18 | 3.7 | 0.350 | 91 | 4.7 | 53 | 3.2 | 0.017 | 115 | 4.5 | 29 | 2.8 | 0.044 |
| **Contacts with rehabilitation teams** | | | | | | | | | | | | | | | |
| 0 | 2830 | 91.6 | 450 | 91.8 | | 1760 | 90.7 | 1520 | 92.7 | | 2321 | 91.4 | 959 | 92.0 | |
| 1–3 | 186 | 6.0 | 29 | 5.9 | | 127 | 6.5 | 88 | 5.4 | | 153 | 6.0 | 62 | 6.0 | |
| ≥4 | 75 | 2.4 | 11 | 2.2 | 0.966 | 54 | 2.8 | 32 | 2.0 | 0.082 | 65 | 2.6 | 21 | 2.0 | 0.622 |
| **Days in hospital** | P value† | | | | | P value† | | | | | P value† | | | | |
| Mean (SD) | 11.85 | (14.15) | 24.98 | (13.32) | | 10.21 | (14.9) | 17.71 | (13.48) | | 13.1 | (15.21) | 14.98 | (13.46) | |
| Median (IQR) | 7 | (0.0–19.0) | 23 | (15.0–32.0) | <0.001 | 2 | (0.0–16.0) | 15 | (8.0–25.0) | <0.001 | 8 | (0.0–20.0) | 11 | (5.0–21.0) | <0.001 |

*$\chi^2$ for trend p value.
†Wilcoxon rank-sum test p value.
ED, emergency department visits; EHA, emergency hospital admissions; GP, general practitioner.

**Table 3** Association between sociodemographic, illness-related and service-related factors with three outcome measures for acute end-of-life care in the last 3 months of life

| | 3 or more EHA last 3 months n=3472 | | | | 1 or more EHA last month n=3441 | | | | 1 or more ED last 2 weeks n=3441 | | | |
|---|---|---|---|---|---|---|---|---|---|---|---|---|
| | Univariate | | Multivariate | | Univariate | | Multivariate | | Univariate | | Multivariate | |
| | RR | 95% CI | RR | 95% CI | RR | 95% CI | RR | 95% CI | RR | 95% CI | RR | 95% CI |
| Age | 0.98 | (0.97 to 0.99) | 0.98 | (0.97 to 0.99) | 0.99 | (0.99 to 0.10) | 0.99 | (0.99 to 1.00) | 0.99 | (0.99 to 1.00) | 0.99 | (0.99 to 1.00) |
| Gender (ref. female) | | | | | | | | | | | | |
| Male | 1.15 | (0.97 to 1.36) | 1.10 | (0.92 to 1.31) | 1.10 | (1.02 to 1.20) | 1.11 | (1.02 to 1.20) | 1.09 | (0.98 to 1.21) | 1.10 | (0.98 to 1.23) |
| IMD quintile (ref. 1, most deprived) | | | | | | | | | | | | |
| 2 | 0.79 | (0.63 to 0.99) | 0.82 | (0.65 to 1.03) | 1.00 | (0.90 to 1.12) | 1.01 | (0.91 to 1.12) | 0.97 | (0.84 to 1.11) | 1.01 | (0.89 to 1.15) |
| 3 | 0.80 | (0.62 to 1.02) | 0.88 | (0.70 to 1.12) | 1.01 | (0.90 to 1.13) | 1.03 | (0.92 to 1.15) | 0.90 | (0.77 to 1.06) | 0.93 | (0.80 to 1.15) |
| 4 | 0.82 | (0.62 to 1.08) | 0.96 | (0.75 to 1.24) | 0.92 | (0.81 to 1.05) | 0.96 | (0.85 to 1.09) | 0.95 | (0.80 to 1.13) | 1.00 | (0.84 to 1.18) |
| 5 | 0.68 | (0.49 to 0.93) | 0.95 | (0.68 to 1.31) | 0.97 | (0.81 to 1.16) | 1.06 | (0.90 to 1.25) | 0.88 | (0.71 to 1.10) | 0.98 | (0.79 to 1.22) |
| Lived in care home (ref. no) | 0.43 | (0.24 to 0.78) | 0.53 | (0.28 to 0.98) | 0.54 | (0.41 to 0.72) | 0.54 | (0.41 to 0.72) | 0.71 | (0.51 to 0.99) | 0.70 | (0.49 to 0.99) |
| Type of cancer (ref. bowel) | | | | | | | | | | | | |
| Lung | 1.53 | (1.13 to 2.06) | 1.60 | (1.16 to 2.20) | 1.11 | (0.98 to 1.25) | 1.08 | (0.96 to 2.23) | 1.04 | (0.88 to 1.22) | 1.01 | (0.85 to 1.19) |
| Prostate | 1.57 | (1.09 to 2.28) | 1.57 | (1.07 to 2.30) | 1.03 | (0.89 to 1.21) | 0.98 | (0.84 to 2.15) | 0.91 | (0.74 to 1.11) | 0.89 | (0.72 to 1.10) |
| Breast | 1.32 | (0.88 to 1.97) | 1.24 | (0.82 to 1.87) | 1.15 | (0.98 to 1.34) | 1.19 | (1.02 to 1.39) | 1.11 | (0.93 to 1.34) | 1.16 | (0.94 to 1.42) |
| Pancreas | 1.34 | (0.88 to 2.04) | 1.23 | (0.93 to 2.08) | 1.05 | (0.89 to 1.25) | 1.02 | (0.87 to 2.20) | 0.85 | (0.67 to 1.09) | 0.82 | (0.63 to 1.06) |
| Haematological | 1.63 | (1.01 to 2.61) | 1.23 | (0.73 to 2.07) | 1.03 | (0.83 to 1.29) | 0.91 | (0.72 to 2.15) | 0.98 | (0.74 to 1.31) | 0.93 | (0.68 to 1.26) |
| Other | 1.24 | (0.91 to 1.69) | 1.15 | (0.83 to 1.58) | 1.00 | (0.88 to 1.12) | 0.96 | (0.86 to 1.08) | 0.85 | (0.73 to 0.99) | 0.82 | (0.70 to 0.96) |
| Number QoF comorbidities (ref. 0) | | | | | | | | | | | | |
| 1 | 0.81 | (0.64 to 1.03) | 0.88 | (0.69 to 1.11) | 0.90 | (0.81 to 1.00) | 0.92 | (0.83 to 1.02) | 0.82 | (0.70 to 0.96) | 0.88 | (0.75 to 1.04) |
| 2 | 0.83 | (0.65 to 1.07) | 0.99 | (0.76 to 1.30) | 0.99 | (0.89 to 1.10) | 1.06 | (0.95 to 1.18) | 0.95 | (0.81 to 1.11) | 1.05 | (0.89 to 1.25) |
| 3 | 1.01 | (0.79 to 1.29) | 1.23 | (0.94 to 1.62) | 0.92 | (0.82 to 1.04) | 0.99 | (0.87 to 1.12) | 0.93 | (0.79 to 1.10) | 1.01 | (0.84 to 1.21) |
| ≥4 | 0.80 | (0.62 to 1.04) | 0.98 | (0.74 to 1.30) | 1.04 | (0.92 to 1.17) | 1.11 | (0.97 to 1.27) | 1.01 | (0.86 to 1.18) | 1.09 | (0.89 to 1.32) |
| QoF comorbidities (ref. no) | | | | | | | | | | | | |
| COPD | 1.09 | (0.88 to 1.34) | * | * | 1.11 | (1.01 to 1.23) | 1.07 | (0.97 to 1.19) | 1.05 | (0.91 to 1.21) | * | * |
| Depression | 1.01 | (0.84 to 1.22) | * | * | 1.04 | (0.94 to 1.14) | * | * | 0.98 | (0.86 to 1.11) | * | * |
| Diabetes | 1.02 | (0.86 to 1.22) | * | * | 1.04 | (0.97 to 1.12) | * | * | 1.09 | (0.97 to 1.22) | * | * |
| Hypertension | 0.92 | (0.79 to 1.07) | * | * | 0.98 | (0.92 to 1.05) | * | * | 0.96 | (0.87 to 1.07) | * | * |
| Dementia | 0.62 | (0.44 to 0.88) | 0.78 | (0.54 to 1.14) | 0.84 | (0.73 to 0.97) | 0.94 | (0.82 to 1.07) | 0.86 | (0.70 to 1.06) | * | * |
| CHD | 0.99 | (0.82 to 1.20) | * | * | 1.08 | (0.99 to 1.18) | * | * | 1.16 | (1.03 to 1.30) | 1.14 | (0.99 to 1.31) |

Continued

**Table 3** Continued

| | 3 or more EHA last 3 months | | | | 1 or more EHA last month | | | | 1 or more ED last 2 weeks | | | |
| --- | --- | --- | --- | --- | --- | --- | --- | --- | --- | --- | --- | --- |
| | n=3472 | | | | n=3441 | | | | n=3441 | | | |
| | Univariate | | Multivariate | | Univariate | | Multivariate | | Univariate | | Multivariate | |
| | RR | 95% CI | RR | 95% CI | RR | 95% CI | RR | 95% CI | RR | 95% CI | RR | 95% CI |
| Contacts with GP practice (ref. 0–3) | | | | | | | | | | | | |
| 4–10 | 1.21 | (1.01 to 1.46) | 1.18 | (0.98 to 1.41) | 1.05 | (0.97 to 1.13) | * | * | 1.12 | (1.00 to 1.24) | 1.10 | (0.98 to 1.22) |
| ≥11 | 1.59 | (1.29 to 1.95) | 1.63 | (1.33 to 1.99) | 1.08 | (0.98 to 1.19) | * | * | 1.19 | (1.03 to 1.38) | 1.27 | (1.10 to 1.47) |
| Contacts with community nurses (ref. 0–3) | | | | | | | | | | | | |
| 4–12 | 1.08 | (0.90 to 1.29) | * | * | 1.04 | (0.96 to 1.13) | 1.06 | (0.98 to 1.15) | 0.97 | (0.86 to 1.10) | 0.96 | (0.85 to 1.08) |
| ≥13 | 1.01 | (0.83 to 1.24) | * | * | 0.84 | (0.76 to 0.93) | 0.88 | (0.90 to 0.98) | 0.80 | (0.69 to 0.93) | 0.79 | (0.68 to 0.92) |
| Contacts with community palliative care teams (ref. 0–3) | | | | | | | | | | | | |
| 4–8 | 0.88 | (0.68 to 1.15) | * | * | 0.91 | (0.80 to 1.04) | 0.95 | (0.82 to 1.08) | 0.97 | (0.82 to 1.14) | 1.01 | (0.85 to 1.21) |
| ≥9 | 0.93 | (0.62 to 1.37) | * | * | 0.78 | (0.63 to 0.96) | 0.85 | (0.69 to 1.04) | 0.70 | (0.50 to 0.97) | 0.78 | (0.56 to 1.08) |
| Contacts with rehabilitation teams (ref. 0) | | | | | | | | | | | | |
| 1–3 | 1.01 | (0.73 to 1.38) | * | * | 0.89 | (0.76 to 1.05) | * | * | 1.01 | (0.82 to 1.25) | * | * |
| ≥4 | 0.93 | (0.55 to 1.58) | * | * | 0.80 | (0.61 to 1.04) | * | * | 0.81 | (0.57 to 1.15) | * | * |
| Days in hospital | 0.94 | (0.94 to 0.95) | 1.04 | (1.03 to 1.04) | 1.02 | (1.01 to 1.02) | 1.02 | (1.01 to 1.02) | 1.00 | (0.99 to 1.00) | 1.00 | (1.00 to 1.01) |

*Variable not included in the model.
CHD, chronic heart disease; COPD, chronic obstructive pulmonary disease; ED, emergency department visit; EHA, emergency hospital admissions; IMD, Index of Multiple Deprivation; QoF, quality of outcomes framework; RR, risk ratio.

important when evaluating quality, they only represent one component of quality at a population level and should be considered alongside other measures of quality such as patient experience and patient-reported outcome measures.

## Strengths and limitations

The Discover dataset holds comprehensive information on healthcare services use from eight different boroughs in London and over 2 million people, including information on primary, community and hospital care. However, our cohort is limited to a London population, which could limit the generalisability of the results.

A limitation of this study is the lack of information on cause of death, time for diagnosis and stage of cancer. Some of the people included might have not died from cancer but from other conditions, and this could vary between different cancer groups. We tried to address this limitation by restricting the sample to people who had been identified as having palliative care needs. However, that approach could have biased the sample towards people with higher or more complex healthcare needs. We derived the date of death from primary care and hospital records, therefore some level of inaccuracy might be expected.[41] Primary care practice contacts were derived from Read codes and therefore it is possible that the number of consultations with the practice was underestimated.[17 18] We excluded administrative contacts and same day records with the primary care practice, as it has been done in other studies, for technical reasons. This approach might underestimate the overall contribution of primary care practices in this study.

We did not have information on the quality of care, continuity, coordination of care or the appropriateness of hospital admissions. Likewise, it was not possible to determine the length of stay or how close to death a person was admitted to the care home facility. These factors could also have an impact on the outcomes of this study. It is likely that some palliative care services were not fully identified, as community palliative care is often provided by the third sector in England, and therefore not consistently included in administrative records.

## CONCLUSIONS

In this population-based cohort study of people with cancer, multiple emergency admissions to hospital in the last 90 days, admissions in the last 30 days and ED visits in the last 2 weeks of life were frequent. Contacts with community nurses were associated with fewer hospital admissions in the last 30 days of life and fewer ED visits in the last 2 weeks of life. More research is needed to explore effective models of end-of-life care in primary care and palliative care integration to address the complexity of the patient population with cancer cared for in primary and community care.

**Contributors** JL and KES had the idea for the study. JL designed the study with input from KES, ZU-H, LAH and IJH. Data analysis was carried out by JL with input from KES, WG, LAH, ZU-H, TN-D, JP and IH. All authors helped to interpret the data.

JL wrote the first draft of the paper. All authors contributed to subsequent drafts and approved the final paper. JL act as the guarantor accepting full responsibility for the work and/or the conduct of the study, had access to the data, and controlled the decision to publish.

**Funding** JL is funded by a Royal Marsden Partners Pan London Research Fellowship Award and the Programa Formacion de Capital Humano Avanzado, Doctorado Becas Chile, 2018 (folio 72190265). KES is funded by a National Institute of Health Research (NIHR) Clinician Scientist Fellowship (CS-2015-15-005) and is the Laing Galazka Chair in Palliative Care at King's College London, funded by an endowment from Cicely Saunders International and the Kirby Laing Foundation. IJH is an NIHR Senior Investigator Emeritus. IJH is supported by the NIHR Applied Research Collaboration South London (NIHR ARC South London) at King's College Hospital NHS Foundation Trust. IJH leads the Palliative and End of Life Care theme of the NIHR ARC South London, and coleads the national theme in this. The views expressed are those of the authors and not necessarily those of the NHS, the NIHR, the Department of Health and Social Care or the funding charities.

**Competing interests** None declared.

**Patient consent for publication** Not required.

**Ethics approval** This study does not involve human participants.

**Provenance and peer review** Not commissioned; externally peer reviewed.

**Data availability statement** Data may be obtained from a third party and are not publicly available. The data that support the findings of this study are available from the Discover data set but restrictions apply to the availability of these data, which were used under license for the current study, and so are not publicly available.

**ORCID iDs**
Javiera Leniz http://orcid.org/0000-0002-9315-4871
Wei Gao http://orcid.org/0000-0001-8298-3415

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
