## [Reviewer comments · BMJ Open]

ARTICLE DETAILS

TITLE (PROVISIONAL)	The association of primary and community care services with emergency visits and hospital admissions at the end of life in people with cancer: a retrospective cohort study.
AUTHORS	Leniz, Javiera; Henson, Lesley; Potter, Jean; Gao, Wei; Newsom-Davis, Tom; Ul-Haq, Zia; Lucas, Amanda; Higginson, Irene; Sleeman, Katherine

VERSION 1 – REVIEW

REVIEWER	Eleanor Kane University of York, Health Sciences
REVIEW RETURNED	03-Jul-2021

GENERAL COMMENTS	The study examines the link between visits to primary care and community services with emergency and hospital admissions at the end of life in people who have cancer. Data from the Discover dataset, a healthcare database of persons resident in North West London which links primary, community and secondary care data, was used. Persons aged 18 or over who had record of cancer in their primary care or hospital notes between 2015 and 2019; who died between 2016 and 2019; and who had a QOF code for palliative care were included. Outcomes were 3 or more emergency admissions in the last 90 days of life; 1 or more emergency admissions in the last 30 days of life; 1 or more emergency department visits in the last 2 weeks of life. Explanatory variables were days primary care consultations in the last 90 days of life; and community care contacts in the last 90 days of life. The authors state that the aims are not only to examine the association between use of primary and community care services with acute hospital use near the end of life for people with cancer, but also to examine how sociodemographic and illness-related factors may relate to the latter. The strength of the study is the primary and community care data; perhaps the sociodemographic factors have been explored elsewhere before. With healthcare patterns often being different among men and women, younger and older, etc, did the authors explore whether the relationships between primary and community care and acute hospital use differed by sociodemographic factors? Stratified analyses by sex and age may help with interpretation of some findings (eg lung and prostate cancer; or care homes) and provide insight of service use within different groups (and could be more informative than the multivariate models where the RRs do not change substantially). Please consider changing the abstract to reflect the use of primary
---

and community care on end of life hospital admission rather than those of older age and in care homes not being admitted to hospital as this is not so surprising. There is evidence here that community care may reduce the need for hospital admission and those in care homes are at lower risk but their need to be admitted is uncertain- as such, the conclusion to the abstract needs to be reconsidered.

Background

End of second paragraph- "this population" is vague and would be better as "cancer patients" or "cancer patients nearing their end of life"

Methods

First sentence "dataset" should be "datasets"

The numbers of organisations feeding into the Discover dataset are listed- how comprehensive is the coverage of all the organisations in North West London?

Persons included were those with cancer...some may have more than one cancer recorded, so how was this dealt with in the cohort selection and grouping of cancer?

Ethics statement is present...may want to add conditions of author's use of data eg depersonalised; use within Discover's trusted research environment,...or similar, as relevant

The use of primary care and community services were categorised based on clinical judgement- it would be useful to give an example or reasoning here? Would also help with understanding what sensitivity analyses were conducted to assess the impact of these categorisations.

Sensitivity analyses were conducted restricted to those who had palliative care recorded in the last 12 months of life. Palliative care can be recorded close to death and too close for hospitalisation to be avoided. How many patients had palliative care recorded in the 2w before death and could a sensitivity analysis be conducted removing these patients?

Results

Page 9, line 11: "More people with lung, prostate and haematological cancers experienced these measures"- need to state the group(s) that lung, prostate and haematological cancer patients are being compared to. If the comparisons are as shown in Table 1, then the authors may like to revisit this statement as it is not necessarily true for each of these cancers across the 3 measures (but is for different cancer groups for some measures).

Similarly, the choice of comorbidities with higher proportions of each measure does not hold for dementia and 3+ admissions in last 90d of life (quite the opposite) and may hold for others eg diabetes.

In light of the above 2 comments, please consider Pearson's chi-squared tests for Table 1- giving statistically significant chi-square tests in the text where appropriate when discussing Table 1 will support the differing distributions (and gives a guide for potential factors to stratify on, or included in multivariate analyses).

Table 2 presents the unadjusted RRs for primary and community care variables' associations with the 3 outcomes, Table 3 the RRs for these variables adjusted for all other variables (as well as the associations with the other factors). The multivariate models are perhaps not showing much difference to what is reported for the unadjusted models; please consider the presentation so it is easier to compare the unadjusted and adjusted variables. Also, please clarify why some cells in Table 3 are blank (e.g. contacts with primary care and +admission in last 30d of life).

	In Table 2 where associations between primary and community care and hospital measures are shown, did the authors consider repeating these analyses excluding those who lived in a care home? Similarly, for Table 3 Age is used as a continuous variable in the multivariate models (Table 3) which makes interpretation tricky. Consider useful categories (eg <65, 65-74, 75-84, 85+, or similar and dependent on numbers) and include frequencies by age groups in Table 1 too. Comparisons are made to the most deprived group...it is more usual to compare to the least deprived group. Please recategorize the IMD quintiles such that group 1 is the most affluent and 5 the most deprived. Multivariate models are adjusted for days in hospital- i.e. the number of days in hospital in the last 90d of life. Given the outcome variables, it is perhaps not surprising that there are associations here. Would replace the findings in Table 3 with those from Model 2 in Tables S1-3. Discussion Please consider where to focus the discussion. It may be better to consider the associations between primary care and community nursing contacts and hospital visits in the last weeks or months of life first as the findings for other factors- living in a care home and younger age- are not that surprising. The second paragraph notes the lower use of secondary care services among those in long-term care in other studies. The inference made is that care home residence prevents the need for acute secondary services; however what is not known, is whether residents of care homes require acute secondary services. The group of care home residents is relatively small and it is not known what the types the care homes were. The authors should moderate this inference both in the discussion and conclusions. Line 37 "in this population"- as before, please clarify which population. Another limitation is the lack of diagnosis date for the cancer- some being close to death, others more distant and possibly not related to their death or care during the period of interest. And this could vary quite substantially between (and within) the different cancer groups.
--	---

REVIEWER	Sarah Mills University of Dundee
REVIEW RETURNED	26-Oct-2021

GENERAL COMMENTS	This is an important subject and a topic that has been historically under-researched. The authors have chosen important matrices for quality of end of life care, and the statistical analysis of these is appropriate and designed to adjust for confounders. Their findings are interesting and useful in practice and policy decision-making for community palliative care. While the analysis in this paper is excellent , more needs to be done to explain how this cohort was chosen and to justify its validity. Without establishing the validity of the cohort group it is impossible to draw any inferences as to universalisability or applicability of the cohort findings. 1. Further information on the completeness of the cohort is required. The 2.6 million patients 'registered with a GP in North West London' doesn't describe how these patients were selected for the cohort, what proportion of the total population they represent or what
--

	measures have been taken to ensure that this population is a representative sample of the total population. 2. The statement that the database is spread across eight CCGs which account for 95% of the population in North West London is unclear - does the cohort represent 95% of the total population, or do those 8 CCGs represent 95% of the total population. Census data suggests the population of North West London is 660,000, so it's unclear where this 2.6 million figure comes from. 3. Please specify which read codes and ICD-10 codes were used for inclusion in the cohort criteria (this can be done in an appendix). 4. Please give more information on what proportion of cancer decedents, who would otherwise have met cohort inclusion criteria, were excluded from this work by restricting it to patients with Palliative Care Read Codes in their notes. Providing that the cohort is a representative sample, then the analysis and conclusions are excellent, important and will have significant implications for policy and practice. The analysis is robust and the use of GEE is an excellent choice to account for associations between multiple attendances per patient. Please give further information on how the model was adjusted in order to take into account number of days spent in hospital in the last 90 days of life. Further information as to how data was anonymised, stored and analysed securely would be beneficial in assessing the ethics of the research. Providing the authors are able to fully justify the cohort selection, and that the cohort is representative and unbiased, then these results are excellent and this article would be an important contribution to a relatively under-published field on this subject.
--	---

VERSION 1 – AUTHOR RESPONSE

Reviewer: 1

Dr. Eleanor Kane, University of York

Comments to the Author:

The study examines the link between visits to primary care and community services with emergency and hospital admissions at the end of life in people who have cancer. Data from the Discover dataset, a healthcare database of persons resident in North West London which links primary, community and secondary care data, was used. Persons aged 18 or over who had record of cancer in their primary care or hospital notes between 2015 and 2019; who died between 2016 and 2019; and who had a QOF code for palliative care were included. Outcomes were 3 or more emergency admissions in the last 90 days of life; 1 or more emergency admissions in the last 30 days of life; 1 or more emergency department visits in the last 2 weeks of life. Explanatory variables were days primary care consultations in the last 90 days of life; and community care contacts in the last 90 days of life.

The authors state that the aims are not only to examine the association between use of primary and community care services with acute hospital use near the end of life for people with cancer, but also to examine how sociodemographic and illness-related factors may relate to the latter.

The strength of the study is the primary and community care data; perhaps the sociodemographic factors have been explored elsewhere before. With healthcare patterns often being different among men and women, younger and older, etc, did the authors explore whether the relationships between primary and community care and acute hospital use differed by sociodemographic factors? Stratified

analyses by sex and age may help with interpretation of some findings (eg lung and prostate cancer; or care homes) and provide insight of service use within different groups (and could be more informative than the multivariate models where the RRs do not change substantially).

Thank you for the very helpful and detailed comments on our manuscript. We agree with the reviewer that the strength of this study is examination of the use of primary and community care services in this population, as sociodemographic characteristics have been explored before.¹⁻³ The aim of this study is to understand the association between primary and community care services and measures of acute hospital use. We realise that our aim, as given at the end of the introduction, was poorly worded in this respect (and not consistent with the aim in the abstract) and we have amended it accordingly as follows:

***'The aim of this study was to describe the association between primary and community care services use with three measures of acute hospital use for people with cancer at the end of life.'* (Page 4)**

The reviewer's helpful comments led us to consider the emphasis on service use and socio-demographic characteristics in our manuscript. We decided to reduce the emphasis on socio-demographic characteristics, so that this does not detract from the focus on primary and community services. We therefore reduced the paragraph in the results where these were reported, and added the following single sentence on sociodemographic characteristics to the section 'Characteristics of the cohort':

***'Older age, white ethnicity and living in a care home were associated with lower chances of all three outcomes (Table 1).'* (Page 8)**

We agree a stratified analysis by sex and age could be a useful addition to the results and could help with interpretation. Unfortunately, our access to the individual level data expired a few months after the submission of this manuscript (March 2021).

Please consider changing the abstract to reflect the use of primary and community care on end of life hospital admission rather than those of older age and in care homes not being admitted to hospital as this is not so surprising. There is evidence here that community care may reduce the need for hospital admission and those in care homes are at lower risk but their need to be admitted is uncertain- as such, the conclusion to the abstract needs to be reconsidered.

Thank you for this important comment, which aligns with our response above. We removed the sentence reporting results for care home residents and added the results for contacts with GP. We also removed from the conclusion in the abstract the comment regarding care homes.

'Results: Of 3581 people, 490 (13.7%) had ≥ 3 admissions in last 90 days, 1640 (45.8%) had ≥ 1 admission in the last 30 days, 1042 (28.6%) had ≥ 1 ED visits in the last 2 weeks; 1069 (29.9%) had more than one of these indicators. Contacts with community nurses in the last three months (≥ 13 vs < 4) was associated with fewer admissions in the last 30 days (RR 0.88, 95% CI 0.79-0.97) and ED visits in the last 2 weeks of life (RR 0.79, 95% CI 0.68-0.92). Contacts with GPs in the last three months (≥ 11 vs < 4) was associated with higher risk of ≥ 3 admissions in the last 90 days (RR 1.63, 95% CI 1.33-1.99) and ED visits in the last 2 weeks of life (RR 1.27, 95% CI 1.10-1.47).

***Conclusions: Expanding community nursing could reduce acute hospital use at the end of life and improve quality of care.'* (Page 2)**

Background

End of second paragraph- "this population" is vague and would be better as "cancer patients" or "cancer patients nearing their end of life"

Thank you for this comment. We changed the sentence "this population" to "cancer patients nearing the end of life" (Page 4)

Methods

First sentence "dataset" should be "datasets"

Thank you for this comment. The dataset is called 'Discover dataset', so we used that name as recommended by data controllers and based on the following the article by Bottle et al 2020.⁴

The numbers of organisations feeding into the Discover dataset are listed- how comprehensive is the coverage of all the organisations in North West London?

Thank you for this important comment. The Discover dataset includes information from 97% of health and social care providers from the NHS in North West London. That includes GP practices, community care service use and social care use, and exclude private and third sector providers, such as hospices services. We added more information in the 'Design and Data sources' section:

***'Of 370 health and social care provider organisations from the National Health Service (NHS) in North West London, 359 (97%) have a data sharing agreement and submit their records to the dataset. Organisations feeding records to the dataset include primary care practices, mental health, community trusts and hospital care attended by North West London patients, and exclude private and third sector providers, such as hospices services.'* (Page 5)**

Persons included were those with cancer...some may have more than one cancer recorded, so how was this dealt with in the cohort selection and grouping of cancer?

Thank for this important comment. For selecting the cohort, we search Read codes in primary care records and ICD-10 codes in hospital records for any record of a diagnosis of cancer between the 1st January 2015 until 2019. People were included if at least one record of a diagnosis of cancer was identified. Read codes and ICD-10 codes used are reported in Box S1 of supplementary material. To define the type of cancer, we used Read codes and ICD-10 codes specific for each type of cancer. Only 6% of the cohort had more than one type of cancer recorded and these were included in the 'Other' category. The Read codes and ICD-10 codes used are also available in Box S1 of supplementary material. We added the following sentence in the population section of the methods:

***'Adults (aged 18 or over) included in the Discover dataset with at least one record of a cancer diagnosis recorded at any point from 1st January 2015 onwards in primary care practice or hospital in-patient records using Read Codes and International Statistical Classification of Diseases and Related Health Problems (ICD) 10 codes respectively (Codes available in Supplemental Box S1).'* (Page 5)**

And the following sentence in the co-variables section:

***'Only 6% of the cohort had more than one cancer recorded and were included in the 'Other' category.'* (Page 6)**

Ethics statement is present...may want to add conditions of author's use of data eg depersonalised; use within Discover's trusted research environment,...or similar, as relevant

Thank you for this relevant comment. We agree that information is relevant to understand the level of security of the data. We added the following paragraph to the Design and Data sources section:

'The Discover dataset is a platform that enables researcher access to pseudonymised patient-level data drawn from the Whole Systems Integrated Care (WSIC) local data warehouse for research purposes. Discover dataset is maintained and interrogated on a secure server and extracts of data are then aggregated in compliance with the Information Governance suppression rule where numbers below 5 are annotated as <5. In this process, the de-identified data is rendered anonymised by stripping out any information that would allow re-identification of an individual's identity. Discover

dataset is accessible via Discover-NOW Health Data Research Hub for Real World Evidence through their data scientist specialists and IG committee-approved analysts, hosted by Imperial College Health Partners.’ (Page 4)

The use of primary care and community services were categorised based on clinical judgement- it would be useful to give an example or reasoning here? Would also help with understanding what sensitivity analyses were conducted to assess the impact of these categorisations.

Thank you for this comment. We categorised primary and community services because the relationship between the log-odds of the outcome and service variables was not linear but also to facilitate interpretation. While other strategies could be used to address this problem, we decided to use a pragmatic approach similar to previous studies.⁵⁻⁷ In consultation with the research team, including clinicians and researchers, we decided to use categories to identify people with one or fewer contacts per month, people between 1 contact per month to 1 contact per week, and people with more than 1 contact per week approximately. In order to achieve the number of people per cell requested by Discover for the data to be extracted from the secure environment, we adapted that approach for community palliative care teams and rehabilitation team contacts. Apologies for not explaining this reasoning in the text. We added the following sentence in the analysis section:

‘To facilitate interpretation, we categorized primary and community care services contacts based on clinical judgement. Categories were approximately one or fewer contacts per month, more than one contact per month but less than one contact per week, and more than one contact per week, depending on the distribution. Because number of contacts with palliative care and rehabilitation teams were small, we adapted these categories.’ (Page 7)

To understand to what extent these categories had an effect on the association of primary and community services, we performed a sensitivity analysis including the variables as continuous. We appreciate this was not clearly explain in the analysis section, so we added the following sentence:

‘(2) to understand the impact of categorization of primary and community care services in the model, we used the same model but with the corresponding primary and community care service use variables as continuous.’ (Page 7)

Sensitivity analyses were conducted restricted to those who had palliative care recorded in the last 12 months of life. Palliative care can be recorded close to death and too close for hospitalisation to be avoided. How many patients had palliative care recorded in the 2w before death and could a sensitivity analysis be conducted removing these patients?

Thank you for this comment. We decided to restrict the cohort to patients who had a record of palliative care need to make sure the cohort represented patients who were considered likely to die in the near future. This was done as we did not have information on the cause of death from the death certificate, or the severity / stage of the cancer. Specifically, we did not aim to explore whether identifying palliative care needs in this population had an impact on hospital admissions. On average, people in the cohort had 3.2 (Median of 2) codes for palliative care QoF recorded in their clinical records. The average number of days from death to the last Palliative Care QoF records was 81.1 days, with a median of 29 (IQR 7-92). Removing people who had palliative care needs identified close to death (eg in the last 2 weeks) would not therefore serve the purpose of this analysis. On the other hand, people with a record of Palliative Care QoF before the last year of life (but not during the last year of life) might represent people who recovered from cancer but died suddenly or from other conditions. We therefore performed a sensitivity analysis removing these patients. Results from that sensitivity analysis do not show different results and are available in the supplementary material.

We apologies if this was not clear in the text. We added the following sentence to the ‘Population’ section:

‘As we did not have information on the cause of death or cancer severity, we restricted our sample to people who had been identified as having palliative care needs in primary care records at any time based on the Quality of Outcomes Framework (QoF) Read Codes for the Palliative Care register,⁸ to include people whose death could be considered expected rather than sudden.’ (Page 5)

Results

Page 9, line 11: “More people with lung, prostate and haematological cancers experienced these measures”- need to state the group(s) that lung, prostate and haematological cancer patients are being compared to. If the comparisons are as shown in Table 1, then the authors may like to revisit this statement as it is not necessarily true for each of these cancers across the 3 measures (but is for different cancer groups for some measures).

Similarly, the choice of comorbidities with higher proportions of each measure does not hold for dementia and 3+ admissions in last 90d of life (quite the opposite) and may hold for others eg diabetes.

Thank you for these important comments. We revised this paragraph substantially to keep the focus on primary and community services as our main explanatory variables. We simplified results regarding socio-demographic characteristics, and moved this to the ‘Characteristics of the cohort section’:

‘Older age, white ethnicity and living in a care home were associated with lower chances of all three outcomes (Table 1).’ (Page 8)

In light of the above 2 comments, please consider Pearson’s chi-squared tests for Table 1- giving statistically significant chi-square tests in the text where appropriate when discussing Table 1 will support the differing distributions (and gives a guide for potential factors to stratify on, or included in multivariate analyses).

Thank you for this comment. We appreciate the sentence used in the analysis section might have been confusing and not adding the p-values in the table did not help. We added the p-values in table 1. To be consistent, we also added the p-values in table 2. We added a sentence in the analysis section mentioning the tests used as follow:

‘A Pearson’ s chi2 test for the trend for categorical variables and t-test and Wilcoxon rank-sum test for age and days in hospital respectively was used to evaluate the association between each variable and the outcomes.’ (Page 7)

To include variables in the multivariate analysis, we forced age, gender, IMD quintile, living in a care home, type of cancer and number of comorbidities as we wanted to adjust for those variables as confounders. We selected specific QoF comorbidities according to results from the univariate analysis to avoid overadjustment, as we were including number of comorbidities as confounder. We selected service use variables as main explanatory variables according results from the univariate analysis. We also changed that sentence in the analysis section for clarify as follow:

‘For the multivariate model, we adjusted by age, gender, IMD quintile, care home residence, type of cancer and number of QoF comorbidities. We selected specific comorbidities, and primary and community care services use according to a priori hypotheses and according to significance in unadjusted analysis (p<=0.05).’ (Page 7)

Table 2 presents the unadjusted RRs for primary and community care variables’ associations with the 3 outcomes, Table 3 the RRs for these variables adjusted for all other variables (as well as the associations with the other factors). The multivariate models are perhaps not showing much difference to what is reported for the unadjusted models; please consider the presentation so it is easier to compare the unadjusted and adjusted variables. Also, please clarify why some cells in Table 3 are blank (e.g. contacts with primary care and +admission in last 30d of life).

Thank you for this suggestion. We restructured both table 2 and 3 to avoid duplicating information. In table 2, we only included the p-values. In table 3 we included the unadjusted and adjusted RR and 95% confidence intervals for all the variables. These changes have made table 3 very populated. We considered removing the RR and confidence intervals for sociodemographic variables and keeping only the main explanatory variables (primary and community service use variables) in the main table (as sociodemographic variables were mainly used as confounders, and could be reported in the supplementary material). On balance, we decided to keep the sociodemographic variables in the table for clarity, but we are happy to move them to the supplementary material if the reviewers think that would be better. Blank spaces in the adjusted columns are because the variable was not included in the multivariate model. We added a note at the end of table 3 explaining that.

In Table 2 where associations between primary and community care and hospital measures are shown, did the authors consider repeating these analyses excluding those who lived in a care home?

Thank you for this interesting suggestion. We did not repeat the analysis excluding people who lived in care homes as the number of people living in care homes was small (<6% of the total sample).

Similarly, for Table 3 Age is used as a continuous variable in the multivariate models (Table 3) which makes interpretation tricky. Consider useful categories (eg <65, 65-74, 75-84, 85+, or similar and dependent on numbers) and include frequencies by age groups in Table 1 too.

We appreciate this comment. We used age as a continuous variable as our main purpose was to adjust the models by age and to avoid the inflation of type I error rate that could occur when continuous confounding variables are categorised for the analysis.⁹

Comparisons are made to the most deprived group...it is more usual to compare to the least deprived group. Please recategorize the IMD quintiles such that group 1 is the most affluent and 5 the most deprived.

Thank you for this comment. We used The English Indices of Deprivation 2015 guidance to report IMD quintiles,¹⁰ which recommend to use 1 for most deprived areas. We believe recategorizing the IMD quintiles would lead to confusion. We added this reference to the manuscript. We agree with the reviewer the least deprived group has been more frequently used as a reference in studies. However, other references groups have been reported as well.^{5, 11-13} Considering we are using IMD quintile as a confounder factor, we do not believe changing the reference category would make a difference to the results. We added the following sentence in the manuscript:

'The 2015 IMD was derived at Lower Super Output Areas (LSOAs) from the patients' last address registered in the system and reported according to The English Indices of Deprivation 2015 guidance.¹⁹' (Page 6)

We also added the sentence '(Most deprived)' and '(Least deprived)' in table 1 and 3 for clarity.

Multivariate models are adjusted for days in hospital- i.e. the number of days in hospital in the last 90d of life. Given the outcome variables, it is perhaps not surprising that there are associations here. Would replace the findings in Table 3 with those from Model 2 in Tables S1-3.

Thank you for this comment. We agree the association between days in hospital and the three outcomes is not surprising. Our intention was not to include days in hospital as an explanatory variable. We adjusted for days in hospital to account for the fact that people who are in hospital cannot have contacts with primary and community care providers. Also, people with many days in hospital might have fewer transitions of care. As days in hospital was highly correlated with the outcomes, we performed a sensitivity analysis to evaluate the impact of including that variable in the model. Removing that variable from the model did not

change the results significantly. However, we believe it is the right thing to adjust for days in hospital. For that reason, we prefer to keep the variable in the main model presented in the results. We apologise if the sentence included in the analysis section was not clear, and as suggested by the second reviewer, we changed that sentence as follow:

'We included the number of days each person spent in hospital in the last 90 days of life as a continuous variable in the models to account for the fact that if someone is in hospital, they cannot receive care in the community' (Page 7)

Discussion

Please consider where to focus the discussion. It may be better to consider the associations between primary care and community nursing contacts and hospital visits in the last weeks or months of life first as the findings for other factors- living in a care home and younger age- are not that surprising.

Thank you for this important comment, which aligns with those above regarding socio-demographic characteristics. We agree the main focus of the paper should be around primary and community service use. We therefore removed the sentence 'We found that being older and living in care homes were consistently associated with lower risk of having all three indicators' in the first paragraph and removed paragraph 4 in the discussion section.

The second paragraph notes the lower use of secondary care services among those in long-term care in other studies. The inference made is that care home residence prevents the need for acute secondary services; however what is not known, is whether residents of care homes require acute secondary services. The group of care home residents is relatively small and it is not known what the types the care homes were. The authors should moderate this inference both in the discussion and conclusions.

Thank you for this relevant comment. We agree our design does not allow us to make strong inferences regarding the association found between living in care homes and use of emergency services. We also agree the number of people living in care homes in this sample was small and we lack information on their health care needs. We therefore changed that paragraph to account for these limitations as follow:

***'People with cancer that were living in care homes before death had a lower risk of hospital admissions and ED visits at the end of life in this cohort. The number of people living in care homes in our sample was small, and we did not have information on the level of health care needs of this group of people. Nevertheless, these findings are consistent with other international studies showing that people living in long-term facilities are less likely to have transitions to hospital regardless of the cause of death.^{14, 15} Long-term facilities are one of the very few care settings in the community providing continuous care including out-of-hours.^{16, 17} More research is needed to understand the mechanisms that explain this association, as well as to explore differences in health care provision and health care needs between people living in care homes and in the community.'* (Page 16)**

Line 37 "in this population"- as before, please clarify which population.

Thank you for this comment. We changed the sentence to:

***'Community nurses have an important role providing physical care, managing symptoms and medications, educating and giving information to patients and families, and coordinating care, and therefore they could play an important role in avoiding unnecessary hospital admissions at the end of life in people with cancer.'* (Page 16)**

Another limitation is the lack of diagnosis date for the cancer- some being close to death, others more distant and possibly not related to their death or care during the period of interest. And this could vary quite substantially between (and within) the different cancer groups.

Thank you for this important comment. We agree the lack of date of diagnosis is a limitation. We restructured that paragraph to include that limitation in the discussion section as follow:

'A limitation of this study is the lack of information on cause of death, time for diagnosis and stage of cancer. Some of these people included might have not died from cancer but from other conditions, and this could vary between different cancer groups. We tried to address this limitation by restricting the sample to people who had been identified as having palliative care needs. However, that approach could have biased the sample toward people with higher or more complex health care needs.'
(Page 18)

Reviewer: 2

Dr. Sarah Mills, University of Dundee, The Mackenzie Building Comments to the Author:
This is an important subject and a topic that has been historically under-researched. The authors have chosen important matrices for quality of end of life care, and the statistical analysis of these is appropriate and designed to adjust for confounders. Their findings are interesting and useful in practice and policy decision-making for community palliative care.

Thank you for these positive comments

While the analysis in this paper is excellent, more needs to be done to explain how this cohort was chosen and to justify its validity. Without establishing the validity of the cohort group it is impossible to draw any inferences as to universalisability or applicability of the cohort findings.

Thank you. We have made changes to the manuscript to explain better the source of the data and the generalizability of the cohort clarify the reviewer's questions. We added some specific comments of the changes made below.

Further information on the completeness of the cohort is required. The 2.6 million patients 'registered with a GP in North West London' doesn't describe how these patients were selected for the cohort, what proportion of the total population they represent or what measures have been taken to ensure that this population is a representative sample of the total population. The statement that the database is spread across eight CCGs which account for 95% of the population in North West London is unclear - does the cohort represent 95% of the total population, or do those 8 CCGs represent 95% of the total population.

Thank you for this important comment and apologies for being unclear. The Discover dataset includes clinical records from all patients enrolled in 365 of 366 general practices (GP) registered in 2018 in the eight boroughs that comprise the North West London STP area (Brent, Ealing, Harrow, Hillingdon, Hounslow, Hammersmith & Fulham, Westminster and Kensington & Chelsea). The number of patients registered in those 365 GP practices represent 95% of all enrolled patients in North West London GP practices. We changed the sentence to:

'This is a retrospective cohort study using the Discover dataset, one of Europe's largest linked longitudinal de-identified datasets that includes over 95% of all patients registered with a general practitioner in North West London.' (Page 4)

Census data suggests the population of North West London is 660,000, so it's unclear where this 2.6 million figure comes from.

Thank you for your comment. According to information from Discover-now, the number of living people in the dataset in June 2019 was 2.37 M. According to the Census 2011, there were 1.9M people living in the 8 boroughs that comprise the North West London STP area (Brent, Ealing, Harrow, Hillingdon, Hounslow, Hammersmith & Fulham, Westminster and Kensington & Chelsea).¹⁸ According to ONS population estimates, the estimated population in June 2019 for the eight boroughs participating in the Discover dataset was 2.1M.¹⁹ The difference could be due to two main reasons: first, the figure from ONS correspond to an estimation from the

2011 Census. Therefore, it is possible that the actual number of residents is higher. Second, it is possible that some patients that do not live in the 8 boroughs are registered in one of the GP practices contributing with data to Discover, and therefore, their primary care records would be included. This phenomenon has been described before and tends to be more marked in London boroughs due to a more transient population. We recognised the original sentence was not sufficiently clear. We changed that section to improve clarity:

'In June 2019, the database held records for a total of 2.37 million patients spread across eight Clinical Commissioning Groups (CCGs). The estimated total population for the eight boroughs contributing data to Discover was 2.1M in mid-2019. Differences in the population estimated and the number of patients in the dataset could be explained by people being enrolled in a GP practice contributing to the dataset but whose usual place of residence is in another area.' (Page 5)

Please specify which read codes and ICD-10 codes were used for inclusion in the cohort criteria (this can be done in an appendix).

We report all our codes, including the ICD-10 codes used for the inclusion in the cohort, in the supplementary material. In case the reviewers cannot access the supplementary material from the submission platform, we are adding a copy at the end of this document. In addition, we added the following sentence to clarify ICD-10 codes are also included in the supplementary material:

"(Codes available in Supplemental Box S1)" (Page 5)

Please give more information on what proportion of cancer decedents, who would otherwise have met cohort inclusion criteria, were excluded from this work by restricting it to patients with Palliative Care Read Codes in their notes.

Thank you for this important suggestion. We found 76% of patients with a diagnosis of cancer and who died between 2016-2019 had a Palliative Care QoF record in the system. We added that information in the supplementary materials' flowchart and we added the following sentence in the text:

'We identified 4933 people with a diagnosis of cancer and who died between 2016 and 2019. 3848 (78.0%) of them had a palliative care QoF record in primary care records. After removing 267 people with invalid dates of death and hospital admissions, 3581 people were included in the analysis (Supplemental Figure S1).' (Page 8)

Providing that the cohort is a representative sample, then the analysis and conclusions are excellent, important and will have significant implications for policy and practice.

The analysis is robust and the use of GEE is an excellent choice to account for associations between multiple attendances per patient.

Thank you for this positive comment

Please give further information on how the model was adjusted in order to take into account number of days spent in hospital in the last 90 days of life.

Thank you for your comment. In order to account for the fact that people who spent days in hospital would not be able to have contacts with community and primary care professionals, and will have lower chances of having transitions to hospital, we included in all models the number of days spent in hospital in the last 90 days of life as a continuous variable. We appreciate the sentence referring to this in the analysis section might not have been clear enough, so we changed it as follows:

‘We included the number of days each person spent in hospital in the last 90 days of life as a continuous variable in the models to account for the fact that if someone is in hospital, they cannot receive care in the community.’ (Page 7)

Further information as to how data was anonymised, stored and analysed securely would be beneficial in assessing the ethics of the research.

Thank you for raising this important question. We added the following paragraph at the beginning of the method section:

‘The Discover dataset is a platform that enables researcher access to pseudonymised patient-level data drawn from the Whole Systems Integrated Care (WSIC) local data warehouse for research purposes. Discover dataset is maintained and interrogated on a secure server and extracts of data are then aggregated in compliance with the Information Governance suppression rule where numbers below 5 are annotated as <5. In this process, the de-identified data is rendered anonymised by stripping out any information that would allow re-identification of an individual’s identity. Discover dataset is accessible via Discover-NOW Health Data Research Hub for Real World Evidence through their data scientist specialists and IG committee-approved analysts, hosted by Imperial College Health Partners.’ (Page 4)

Providing the authors are able to fully justify the cohort selection, and that the cohort is representative and unbiased, then these results are excellent and this article would be an important contribution to a relatively under-published field on this subject.

We really appreciate this supportive comment and hope the changes made explain better the level of representativeness of the sample used in this study.

VERSION 2 – REVIEW

REVIEWER	Eleanor Kane University of York, Health Sciences
REVIEW RETURNED	03-Jan-2022
GENERAL COMMENTS	Thank you for addressing my comments.